

# Bacterial communities associated with cell phones and shoes

David A. Coil[1], Russell Y. Neches[1], Jenna M. Lang[1], Guillaume Jospin[1], Wendy E. Brown[2,3], Darlene Cavalier[3,4], Jarrad Hampton-Marcell[5], Jack A. Gilbert[6] and Jonathan A. Eisen[7]

[1] Genome Center, University of California, Davis, CA, United States of America
[2] Department of Biomedical Engineering, University of California, Irvine, CA, United States of America
[3] Science Cheerleaders, Inc., Philadelphia, PA, United States of America
[4] SciStarter.org, Philadelphia, PA, United States of America
[5] Argonne National Laboratory, University of Chicago, Lemont, IL, United States of America
[6] Department of Pediatrics and Scripps Institution of Oceanography, UC San Diego School of Medicine, San Diego, CA, United States of America
[7] Genome Center, Department of Evolution and Ecology, Department of Medical Microbiology and Immunology, University of California, Davis, Davis, CA, United States of America

Corresponding authors
David A. Coil, dcoil@ucdavis.edu
Jonathan A. Eisen,
jaeisen@ucdavis.edu

## ABSTRACT

**Background**. Every human being carries with them a collection of microbes, a collection that is likely both unique to that person, but also dynamic as a result of significant flux with the surrounding environment. The interaction of the human microbiome (i.e., the microbes that are found directly in contact with a person in places such as the gut, mouth, and skin) and the microbiome of accessory objects (e.g., shoes, clothing, phones, jewelry) is of potential interest to both epidemiology and the developing field of microbial forensics. Therefore, the microbiome of personal accessories are of interest because they serve as both a microbial source and sink for an individual, they may provide information about the microbial exposure experienced by an individual, and they can be sampled non-invasively.

**Findings**. We report here a large-scale study of the microbiome found on cell phones and shoes. Cell phones serve as a potential source and sink for skin and oral microbiome, while shoes can act as sampling devices for microbial environmental experience. Using 16S rRNA gene sequencing, we characterized the microbiome of thousands of paired sets of cell phones and shoes from individuals at sporting events, museums, and other venues around the United States.

**Conclusions**. We place this data in the context of previous studies and demonstrate that the microbiome of phones and shoes are different. This difference is driven largely by the presence of "environmental" taxa (taxa from groups that tend to be found in places like soil) on shoes and human-associated taxa (taxa from groups that are abundant in the human microbiome) on phones. This large dataset also contains many novel taxa, highlighting the fact that much of microbial diversity remains uncharacterized, even on commonplace objects.

## INTRODUCTION

Our understanding of the human microbiome (e.g., *McDonald et al., 2018*), the microbiome of the built environment around us (e.g., *National Academies of Sciences, Engineering, and Medicine et al., 2017*), and the interactions between the two (e.g., *Leung & Lee, 2016*) have dramatically expanded in recent years. This understanding has implications for fields ranging from medicine to forensics to architecture. In addition to the millions of microbes that we carry around each day, the majority of people on the planet are thought to now possess a cell phone. Previous work on the microbiome associated with phones has shown that people share a much greater percentage of their microbes with their own phone than with the phones of others (*Meadow, Altrichter & Green, 2014*). Additionally, these authors showed a high correlation between the index finger specifically, and the surface of the owner's phone. As for the environment around us, shoes (or other foot coverings) can act as microbial sampling devices. We have previously described data suggesting this to be the case, as well as demonstrated that the microbiome of cell phones and shoes from the same person are quite distinct (*Lax et al., 2015*).

Though the existence of microbes has been known for a few hundred years, only in recent decades have we come to learn of the existence of the many microorganisms on the planet that have not yet been cultivated in a lab. This so-called "microbial dark matter" (MDM) is understudied and probably makes up a majority of microbial life on the planet (*Solden, Lloyd & Wrighton, 2016*; *Bernard et al., 2018*; *Lloyd et al., 2018*). Sometimes the term refers to any uncultured taxa, while others use it to refer to major evolutionary lineages for which few or no representatives have ever been grown in the lab or studied in detail (*Rinke et al., 2013*). Here we use "MDM" in the latter, more general sense. While MDM taxa are probably best known from extreme environments like acid mine drainage and the deep sea, there are presumably also many relatively unknown taxa on items as commonplace as phones and shoes.

Throughout 2013–2014, we organized public events around the United States for the purpose of swabbing surfaces of the built environment and collecting bacteria for isolation via culturing. Cultured isolates from these samples were screened and a subset of them were sent to the International Space Station (ISS) for growth in microgravity (*Coil et al., 2016*). As part of the public outreach component of this project, we engaged the public in helping collect these swabs, as well as in swabbing their cell phones and shoes for a nationwide microbial biogeography study. Thousands of people participated in this project, and we initially collected ~3,500 paired cell phone/shoe samples, of which we sequenced ~2,500 samples. The intent of examining bacteria on cell phones and shoes was threefold; firstly to scale up the results of previous studies on shoes and phones and to look for patterns in the biogeography at a national scale. The second was to engage people in thinking about cell phones as being a putative proxy for sampling the microbes found on a person and their shoes as being a putative proxy for sampling the microbes found in a person's environment. Lastly, we wanted to search for MDM taxa on common, human-associated objects. To our knowledge, this represents the largest collection of bacterial community sequencing data associated with cell phones or shoes.

## MATERIAL AND METHODS

### Sample collection

Cell phone and shoe samples were collected on sterile cotton swabs (Puritan cotton tipped #25-806) and participants were instructed to "swab for about 15 s as if trying to clean the object". Swabs were kept at room temperature by necessity and then sent overnight to the University of Chicago, where they were kept at −80 °C until processing. Metadata for the samples included the physical location (GPS coordinates), date of sampling, rough age of participants, sample object type (cell phone or shoe), and event (basketball game, museum visit, etc.). Participant age was estimated by the event organizers as primarily children (e.g., an elementary school), primarily adults (e.g., a conference), or a mix (e.g., a baseball game). This study was performed under an expedited review and waiver through the University of Chicago IRB under protocol 'Phones and Shoes Study' IRB13-1091 awarded to Jack Gilbert.

### DNA extraction and sequencing

DNA extractions, library preparation, and Illumina sequencing (paired-end 150 bp) were performed exactly as described in our previous work using swabs from the ISS (*Lang et al., 2017*). In brief: samples were prepared using Mo BIO UltraClean kits, DNA extracted using Zymo ZR-96 kits, DNA amplified using EMP barcoded primer sets targeting the V4 region of the 16S rRNA gene, amplicons were cleaned and pooled and sequenced on an Illumina MiSeq platform.

### Data processing, validation and generation of ASV tables

The dataset (2,486 sequenced samples) was prepared by following the DADA2 protocol ("big data") (*Callahan et al., 2016a*) to generate amplicon sequence variants (ASVs). Each sequencing lane was also pre-processed individually to account for error patterns from different runs or machines. Reads longer than 150 base pairs (bp) were trimmed down to 150 bp before processing with DADA2. Low quality regions of reads were removed by trimming bases that did not satisfy a Q2 quality score. The reads were also trimmed down to a length of 145 bp. Reads containing Ns were discarded and we used two expected errors to filter the overall quality of the read (rather than averaging quality scores) (*Edgar & Flyvbjerg, 2015*). Only forward reads were considered for this study, in order to be consistent with previous work. Quality filtering resulted in 2,230 samples being analyzed. 186,334 unique ASVs were identified and taxonomic assignments were made for these ASVs using the Silva NR v132 database. Samples without complete metadata were excluded. Using Phyloseq, the non-bacterial ASVs that were assigned to mitochondria or chloroplasts (in total 63,838 or 34% of the ASVs) were excluded from further analysis, resulting in 148,535 remaining ASVs. The ASV based filtration reduced the total number of samples to 2,230 (since some samples did not contain any of these final ASVs). After rarefying to 10,000 reads per sample, 44,897 ASVs were no longer represented in the data set and 348 samples were removed due to insufficient ASVs. In total, 17,550,000 of the initial reads were used for the Alpha diversity analyses. The data was additionally filtered to only include ASVs present in >5% of the samples and rarefied again to 10,000 reads per samples which resulted in 2,253

ASVs for 1,672 samples. This version was used in the beta diversity analyses. For Alpha diversity, additional filtering was required, pairing 637 phone samples to shoe samples (totalling 1,274). For the biogeography analysis, samples were summed by Event which resulted in the exclusion of five events, resulting in 34 Event locations being evaluated after the previous pairing step. A table of all filtering/processing steps can be found as Table S1.

## Diversity analyses (alpha, beta, taxonomic, phylogenetic)

Alignment of the observed sequences was performed using Clustal Omega (*Goujon et al., 2010*; *Sievers et al., 2011*), and an approximate maximum likelihood phylogeny was constructed using FastTree2 (*Price, Dehal & Arkin, 2009*; *Price, Dehal & Arkin, 2010*). Metadata was loaded from the mapping files and relevant columns were extracted using Pandas (*McKinney, 2010*) (retained values were: Age, City, Date, Event, Run, Sample, Sport, State, Type). ASV filtering, taxonomic agglomeration, and ordination was performed using phyloseq (*McMurdie & Holmes, 2013*) using Callahan et al. as a guide (*Callahan et al., 2016b*).

The alpha diversity metrics were calculated using phyloseq and ggplot R packages as well as a reduced dataset in which we removed all "Sample: Unknown" samples. We then rarified the samples to 10,000 reads. Only the samples which had corresponding phone and shoe pairs were considered for plotting the Shannon and Observed diversity metrics. We chose 10,000 reads for a rarefaction cut off by plotting all sample's rarefaction curves and picked a cutoff that would balance sample inclusion and enough ASVs. The PCoA ordination of the Bray-Curtis dissimilarity of the ASV data was generated using the ordinate and plot_ordination functions from Phyloseq. As input to the ordination function, we further filtered the ASVs to those represented in at least 5% of the samples then rarefied to 10,000 reads per samples. We exported the ordination coordinates and averaged values for cell phones and shoes separately to find the centroid of the two data spreads. We plotted a line bisecting perpendicularly the segment between the two centroids to highlight the separation between the two groups. We used ggplot2 to overlay this line on the sample and taxa (at the phylum level) versions of the PCoA (*Wickham, 2010*). We ran an ANalysis Of SIMilarity (ANOSIM) test available through the vegan R package to assess the similarities between the phone and shoe samples using Bray-Curtis dissimilarity and 999 permutations (*Oksanen et al., 2011*).

We plotted (ggplot2) the Bray-Curtis dissimilarity (vegan) on the ASV counts of the samples summed by sampling sites (phyloseq) against the physical distances between the sites. We used a custom perl script to calculate the geographical distances using GPS coordinates treating the Earth as a sphere. Mantel tests were done using the ade4 (*Dray & Dufour, 2007*) R package.

## Attribute importance analysis

Random forest and related analyses were done using the scikit-learn v0.21.2 Python package (*Pedregosa et al., 2011*). Variable importance measures were estimated by first training the random forest classifier (*Breiman, 2001*; *Geurts, Ernst & Wehenkel, 2006*; *Pedregosa et al., 2011*) on the final ASV counts and then extracting the attribute importance values, also

called the gini importances or mean decrease impurity (Breiman et al., 1984) from the trained classifiers (*Janitza, Strobl & Boulesteix, 2013*). Other than specifying 50 estimators, the default parameters were used. The figure was generated using the matplotlib Python package (*Hunter, 2007*).

## RESULTS/DISCUSSION

### Alpha diversity

In total, ~3,500 swabs were collected for this study at 38 events (see Table S2 for details on events). Of these, some samples were lost in transit and a further 864 samples were excluded from sequencing due to an irretrievable loss of the sample ID data (computer failure). The exact number of actual swabs originally collected/lost is unknown, due to the distributed nature of the collection as part of a citizen science project. Sequencing was performed on 2,486 samples with 599,386,254 paired end reads generated across four lanes of Illumina HiSeq PE150.

To examine the alpha diversity of these samples, we examined all pairs of samples where both the cell phone and the shoe had at least 10,000 reads. The plot of both observed counts and the Shannon diversity index ($H$) can be seen in Fig. 1. By either measure, shoes have a significantly higher alpha diversity than phones. This is concordant with previous results and presumably results from the greater variety of environmental taxa that shoes might encounter over time.

### Attribute importance

As a method for examining the potential importance of the metadata variables (sample type, sport, location, and sequencing run), we utilized variable importance measures (VIMs). These VIMs were estimated by training a random forest classifier (*Breiman, 2001*; *Geurts, Ernst & Wehenkel, 2006*; *Pedregosa et al., 2011*) to assign samples to their metadata categories (sample type, city, state, sequencing run, and sport) based on their ASV counts, and extracting the variable importance values (*Breiman et.al, 1984*) from the trained classifiers (*Janitza, Strobl & Boulesteix, 2013*). VIMs are implemented as the total decrease in node impurity, weighted by the probability of reaching that node as approximated by the proportion of samples reaching that node, averaged over all trees in the ensemble (https://stackoverflow.com/questions/15810339/how-are-feature-importances-in-randomforestclassifier-determined). Note that variable importance analysis is a distinct application of random forests from the more widely-used classification application. Extracting VIMs does not include the optimization and benchmarking steps required to use random forests in their predictive capacity. Sample feature importances indicate that the sample type (shoe or phone) was the most predictive of the observed community structure, followed by the geographic location of the sample (Fig. S1). The sport played at the venue where the sample was collected is less predictive of the community structure than the sequencing run. Overall, these results support and extend our previous findings that the microbiomes of shoes and phones are distinct. Interestingly, the city where an event took place was more predictive of community structure than state, suggesting the possibility that there are local biogeography effects in patterning the microbial community.

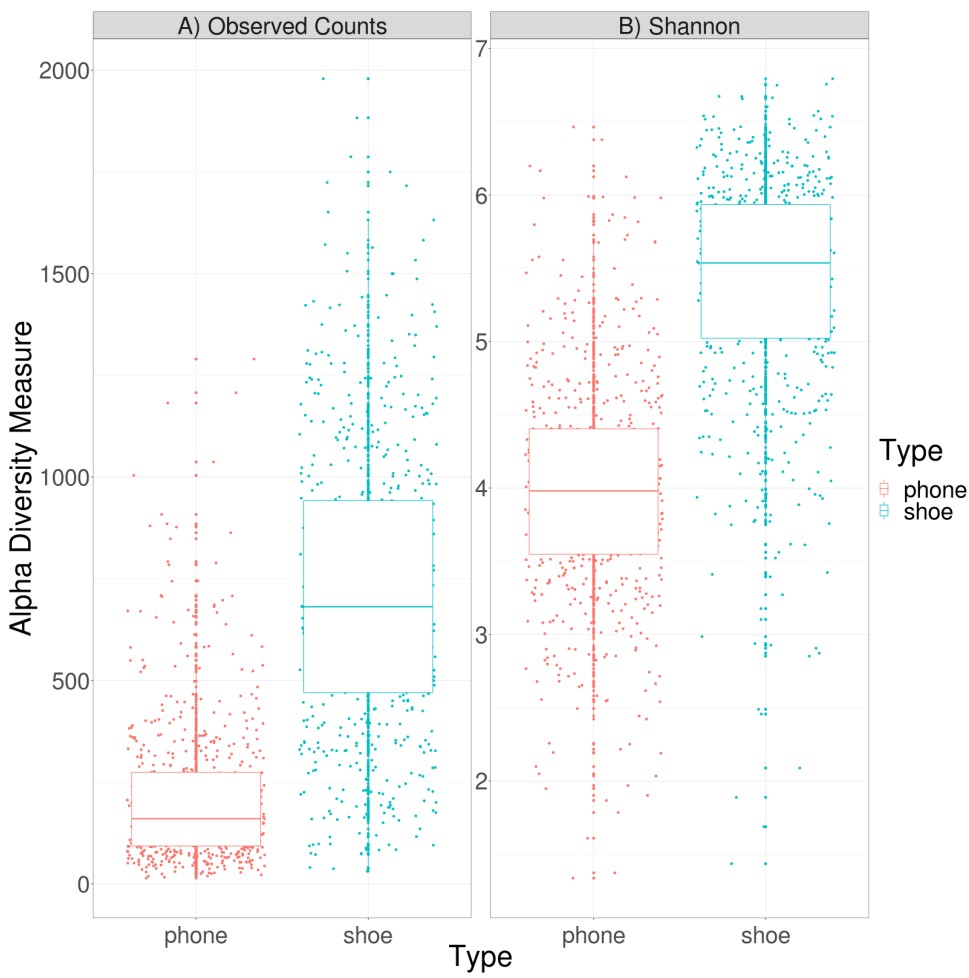

**Figure 1** Alpha diversity of cell phone and shoe samples, calculated by either observed counts (A) or by the Shannon diversity index (B).

## Beta diversity

In order to examine and visualize differences between samples, we plotted a PCoA ordination of samples based on sample to sample Bray-Curtis dissimilarity of the rarefied microbial communities that appear in more than 5% of the samples (Fig. 2). A quick examination of the plot revealed that cell phones (green) and shoes (black) appear to group separately (something seen in prior studies); this is supported by statistical analysis (ANOSIM $R = 0.5736$, $p = .001$).

To further examine the differences between cell phones and shoes, we identified the centroids of the two data spreads (Fig. 2). The line in this figure represents the bisection of these two centroids, to highlight their separation. We then used this bisection line to examine in more detail the taxa that contribute to the separation of shoe and phone samples. We did this by generating a series of plots showing only the ASVs belonging to each phylum separately (Fig. 3), showing only those that were significant in our ANCOM analysis. The line in each plot is the same as in the sample plot in Fig. 2 and those ASVs to
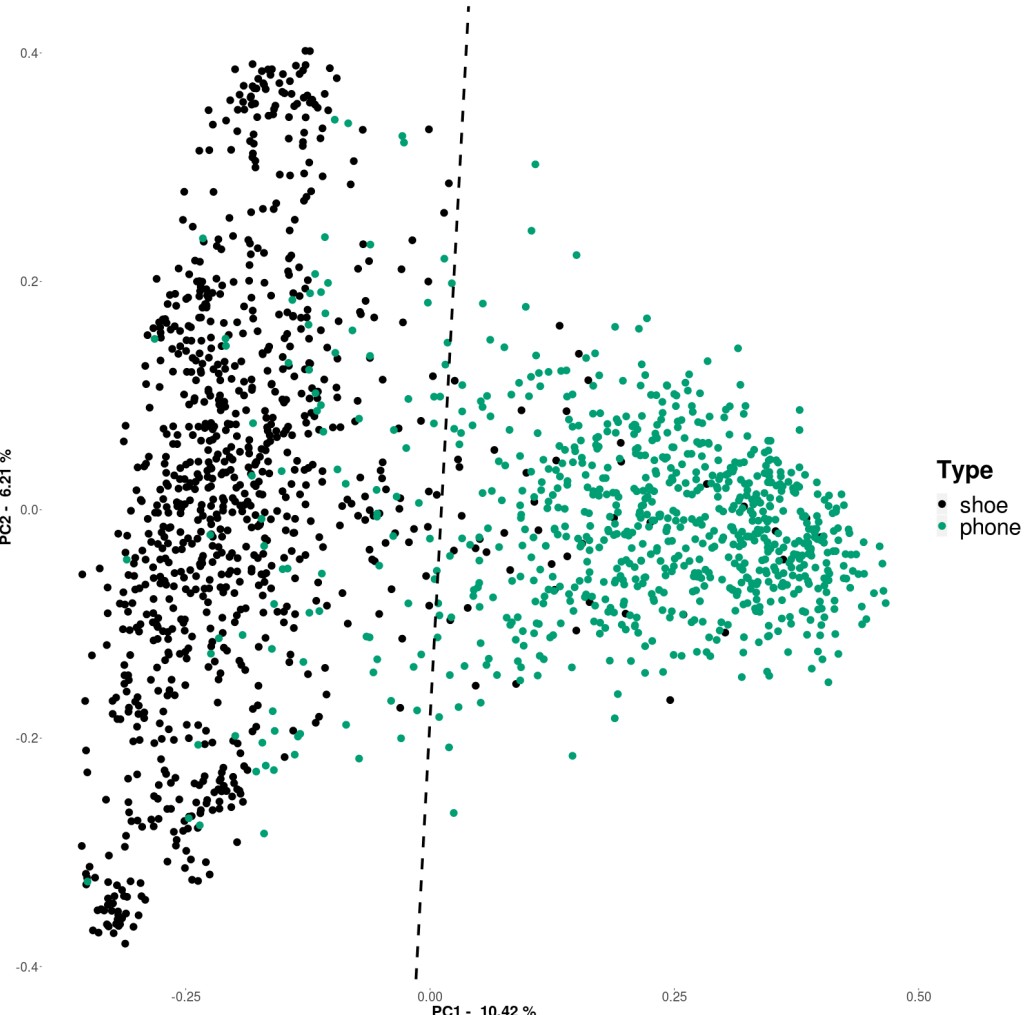

**Figure 2** Principal coordinate (PCoA) analysis plot of Bray–Curtis distances (based on 16S rRNA gene sequence based ASVs, rarefied to 10,000 sequences) for cell phone and shoe samples, colored by sample origin The line is the bisection of the centroids of the two sample types (phones and shoes).

the top/left can be considered to be driving the "phone" portion of the PCoA and the ASVs to the bottom/right can be considered to drive the "shoe" portion of the PCoA. These plots (and the underlying data) show some interesting phyla-specific patterns. Some phyla (e.g., Bacteroides and Firmicutes) have many ASVs on both sides of the line, indicating that there are ASVs from these phyla that are significantly biased towards shoes and others that are significantly biased towards phones.

One phylum (Fusobacteria) contains only ASVs that are skewed towards phones. We believe this is likely due to these ASVs being human associated taxa. For example, the taxonomic assignments of the Fusobacteria ASVs were *Leptotrichia* ($n = 2$) and *Fusobacterium* ($n = 1$); these two genera are generally found in animal microbiomes including the oral microbiome of humans and other mammals (*Eribe & Olsen, 2008*; *Whitman et al., 2015a*; *Whitman et al., 2015b*). On the other hand, there are two phyla

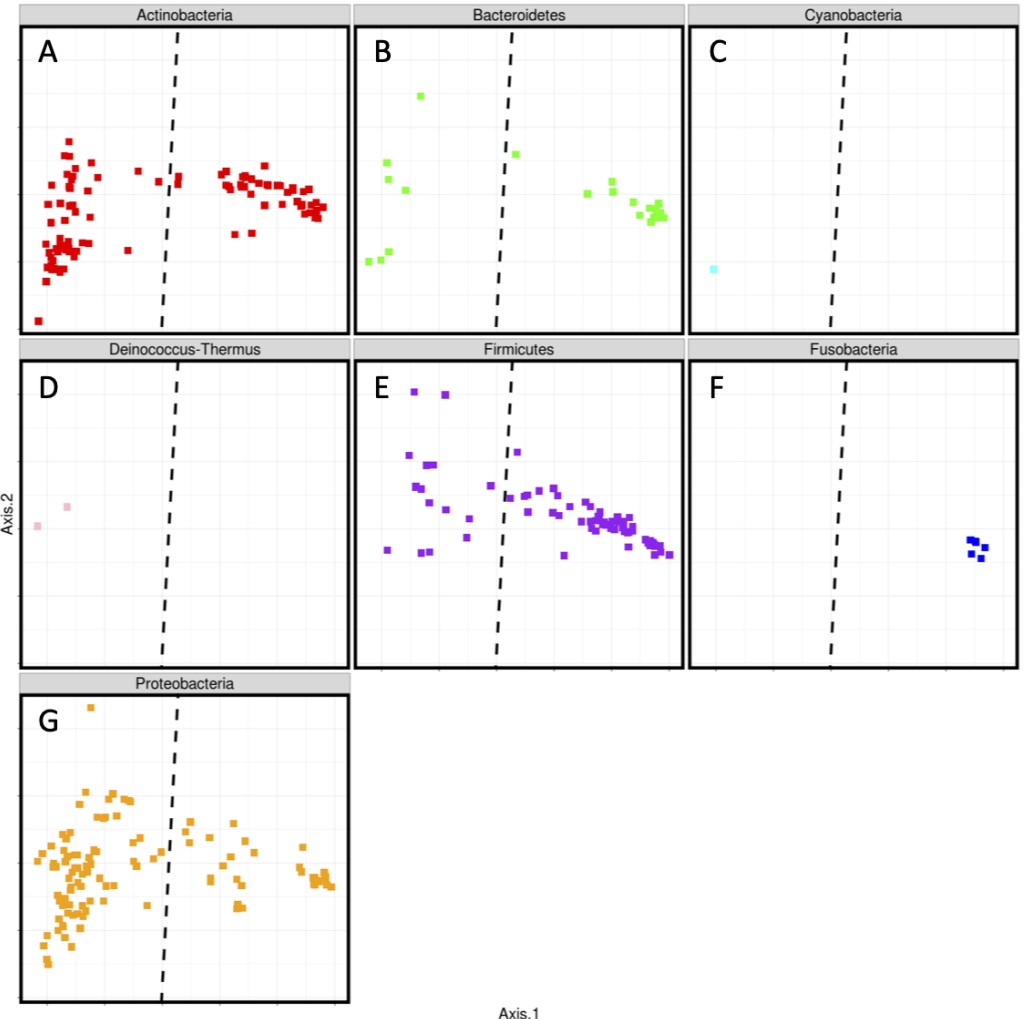

**Figure 3** Split Phyla representation of PCoA ordination of Bray-Curtis dissimilarity of rarefied ASV counts. (A) Actinobacteria; (B) Bacteroidetes; (C) Cyanobacteria; (D) Deinococcus-Thermus; (E) Firmicutes; (F) Fusobacteria; (G) Proteobacteria. Only ANCOM detected, significant ASVs are represented. ASVs biased toward shoes are on the left, those biased towards phones are on the right.

(Deinococcus-Thermus, Cyanobacteria) which include only ASVs that are skewed towards shoes. We presume that these ASVs from these phyla represent taxa from the broader environment (e.g., soil) that would be picked up by shoes. Examination of the taxonomic assignments for these ASVs supports this possibility, with genera assignments including taxa commonly found in water or soil such as *Chroococcidiopsis* e.g., (*Billi et al., 2000*), *Oscillatoria* e.g., (*Carpenter & Price, 1976*), *Truepera* e.g., (*Albuquerque et al., 2005*), and *Deinococcus* e.g., (*Battista, Earl & Park, 1999*).

## Biogeography
This study included sampling sites as close together as within the same city (e.g., multiple events in Philadelphia, PA) as well as sites spread out across the United States. Previous
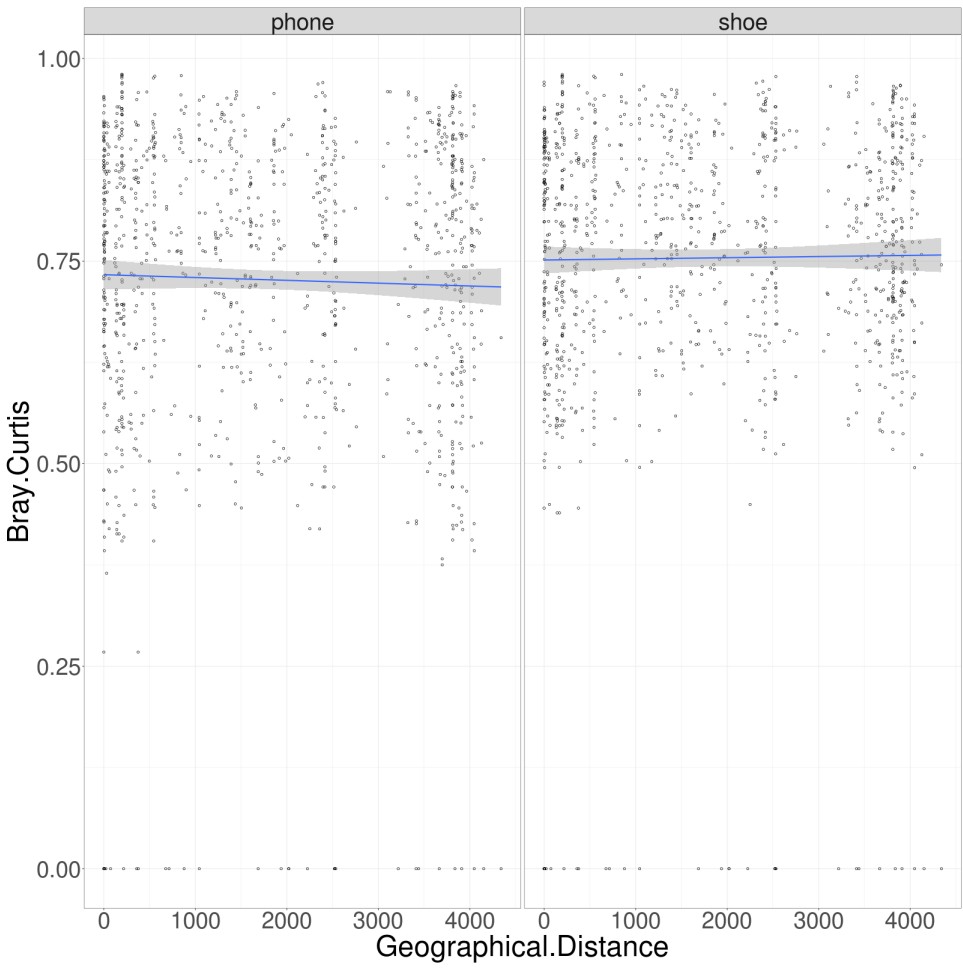

**Figure 4** **Plot of geographic distance in miles versus Bray–Curtis dissimilarity of all pairs of locations, separated by cell phones and shoes.** A Mantel test performed on both the data from cell phones and shoes, comparing the geographic distance to the Bray Curtis distance, showed no correlation (simulated $p$-values of .027 and .005, respectively).

biogeography work on a continental scale (China) showed that environmental bacteria had a strong relationship between community similarity and geographic distance, while Archaea showed no such pattern (*Ma et al., 2017*). We conducted a similar analysis, treating both cell phone and phone samples separately (Fig. 4). Both cell phones and shoes are very "noisy" in this analysis, some samples that are within the same city have radically different communities and some samples thousands of miles apart have very similar bacterial communities. Therefore, we do not observe a significant correlation between community similarity and geographic distance, in either cell phones or shoes.

## Novel evolutionary lineages

Additionally, we examined how many (if any) of these microbes present on cell phones and shoes were from any of the so-called "microbial dark matter" branches in the tree of life. The term "microbial dark matter" or MDM for short is used in this context to refer to

major evolutionary lineages for which few or no representatives have ever been grown in the lab or studied in detail (*Rinke et al., 2013*).

To identify MDM in our data, we searched through the taxonomic annotation of ASVs for those assigned to phyla or candidate phyla which are generally viewed as MDM lineages. Specifically, we considered ASVs assigned to the following groups as being MDM: Aegiribacteria, AncK6, Armatimonadetes, Atribacteria, BRC1, Caldiserica, Calditrichaeota, Chrysiogenetes, Cloacimonetes, Coprothermobacteraeota, Dadabacteria, Dependentiae, Diapherotrites, Edwardsbacteria, Elusimicrobia, Entotheonellaeota, Fervidibacteria, FCPU426, GAL15, Hydrogenedentes, Latescibacteria, Margulisbacteria, Nanoarchaeaeota, Nitrospinae, Omnitrophicaeota, Patescibacteria, PAUC34f, Rokubacteria, RsaHf231, WOR-1, WPS-2, WS1, WS2, WS4, and Zixibacteria. We chose these groups because of all the phyla to which our ASVs were assigned, these are the groups that either contain no cultured representatives or for which most of the phylogenetic diversity within the group is only represented by uncultured taxa. We also then examined the distribution patterns of these ASVs across samples and whether they showed any skew between phones and shoes (Table S3).

This analysis of ASVs assigned to MDM lineages revealed that, in fact, quite a large number of ASVs found in our study were from such MDM groups. In some cases, these ASVs assigned to these groups are quite rare—for example, ASVs from WOR-1, Edwardsbacteria, and Diapherotrites were found to be present in one sample each. However, some were present in a much wider range of samples, and we focused most of our attention on those (Table S3). Of the nine MDM phyla for which ASVs were found to be present in at least 10% of samples (Armatimonadetes, Patescibacteriam, WPS-2, Entotheonellaeota, Dependentiae, BRC1, Rokubacteria, Latescibacteria, Elusimicrobia), all were found more often in shoe samples than phone samples. This is not surprising given that (1) phone samples tend to be enriched for human associated microbes, only a few of which are in current MDM groups and (2) many MDM lineages are known to be found in soil, which is presumably abundant on shoes. Two of these widespread MDM groups (Armatimonadetes, Patescibacteria) were found to have ASVs present in almost 50% of samples. The Armatimonadetes phyla is known to be both diverse and widespread, with soil contributing the most members of this group of any single environment (*Lee, Dunfield & Stott, 2014*). The proposed Patescibacteria superphylum also contains a wide variety of diverse taxa, but the majority are associated with aquatic or semi-aquatic environments (*Sánchez-Osuna, Barbé & Erill, 2017*). Twelve classes and thirteen orders were found to be present in more than 10% of samples. Of these, all were skewed towards shoe samples, except two taxa (Gracilibacteria within Patescibacteria and Absconditabacteriales within Gracilibacteria).

Overall, these results show that, while MDM might be thought of as coming from remote, isolated, or extreme environments, a remarkable fraction of people are traveling around with representatives from these uncultured groups on commonplace objects. This highlights how much we still have to learn about the microbial world around us.

# PeerJ

## SUMMARY

These data support previous work by ourselves and others demonstrating that the microbiome of cell phones and shoes are distinct, even when belonging to the same person. The taxonomic diversity of shoes appears to be much higher than that of phones. In this analysis, we also highlight which phyla are most responsible for the observed differences in microbial communities between phones and shoes. This difference is driven largely by the presence of "environmental" taxa (taxa from groups that tend to be found in places like soil) on shoes and human-associated taxa (taxa from groups that are abundant in the human microbiome) on phones. We did not observe a correlation between geographic distance and community similarity. Lastly, we show that a number of "microbial dark matter" taxa are present, even abundant, on these commonplace objects.

## ACKNOWLEDGEMENTS

The authors would like to thank the many Science Cheerleaders who ran the sampling events. In particular, we'd like to thank Bart Leahy who coordinated the coordinators. We would also like to thank SciStarter and Science Cheerleaders, Inc. for being partners in this work. Thanks to Bretwood Higman for the script used to determine geographic distances. Finally, we would like to thank the thousands of people who gave willingly of their time and their samples as part of participating in this study.

### Funding

Funding for this study was provided by the Alfred P. Sloan Foundation through their program in the "Microbiology of the Built Environment". The funders had no role in study design, data collection and analysis, decision to publish, or preparation of the manuscript.

### Grant Disclosures

The following grant information was disclosed by the authors:
Foundation through their program in the "Microbiology of the Built Environment".

### Competing Interests

Jonathan Eisen is currently on the Scientific Advisory Board of Zymo Research but was not when the work described here was carried out. Jonathan Eisen is an Academic Editor for PeerJ. Darlene Cavalier is the founder and a member of both Science Cheerleader and SciStarter.org. Wendy Brown is a volunteer member of Science Cheerleader. All authors declare that they all use cell phones and wear shoes, but have no other competing interests.

### Author Contributions

- David A. Coil and Russell Y. Neches conceived and designed the experiments, performed the experiments, analyzed the data, prepared figures and/or tables, authored or reviewed drafts of the paper, and approved the final draft.

- Jenna M. Lang, Wendy E. Brown and Darlene Cavalier conceived and designed the experiments, performed the experiments, authored or reviewed drafts of the paper, and approved the final draft.
- Guillaume Jospin analyzed the data, prepared figures and/or tables, authored or reviewed drafts of the paper, and approved the final draft.
- Jarrad Hampton-Marcell and Jack A. Gilbert performed the experiments, authored or reviewed drafts of the paper, and approved the final draft.
- Jonathan A. Eisen conceived and designed the experiments, analyzed the data, prepared figures and/or tables, authored or reviewed drafts of the paper, and approved the final draft.

## Human Ethics

The following information was supplied relating to ethical approvals (i.e., approving body and any reference numbers):

This study was performed under an expedited review and waiver through the University of Chicago IRB under protocol 'Phones and Shoes Study' IRB13-1091 awarded to Jack Gilbert.

## DNA Deposition

The following information was supplied regarding the deposition of DNA sequences:

All raw sequencing data are available at NCBI: PRJNA470730.

https://www.ncbi.nlm.nih.gov/sra/SRP145522.

## Data Availability

All data analysis, metadata, supporting files and intermediate analysis files are available at Zenodo:

Guillaume Jospin, & Russell Neches (2020, February 13). ryneches/shoephone: Community analysis of cell phones and shoes : Project MERCCURI (Version 1.1). Zenodo. http://doi.org/10.5281/zenodo.3665855.

An interactive visualization of this data is available at http://www.phinch.org. A website describing this project and related ones can be found at http://www.spacemicrobes.org.

## Supplemental Information

Supplemental information for this article can be found online at http://dx.doi.org/10.7717/peerj.9235#supplemental-information.

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
