# Peer review of "Bacterial communities associated with cell phones and shoes"

_PeerJ, doi:10.7717/peerj.9235_

## Round 0.1 · original submission · Major Revisions

Three peer reviewers made extensive comments that I think will help to improve the paper substantially. I do apologize for the delay: it was difficult to find reviewers that were not connected to your very collaborative group! All agreed that there is interest in the microbiome of the built environment. However, there appears to a mismatch between the intended purpose of the study and the results as presented.

All three reviewers found room for improvement in the basic reporting, presentation of the study design, and validity of the findings. I also agree that the authors should limit the results section to the major findings, and move the data curation information to a separate section. While I can see how curation of the final, quality controlled data set might be considered results, it detracts from the main message of the paper.

Finally, though it is your prerogative to wish for the analysis as presented to be reviewed, I hope you will consider doing some of the additional analyses that are suggested.

Reviewer 1 ·

Basic reporting

1.1 The manuscript suffers from occasional typos. Please check throughout manuscript carefully.

1.2 The authors should pay attention on sections of the text that may be more suited in other sections of the manuscript.

1.3 I feel that because a lot of the text currently in the results section should in fact be in the Intro or Materials sections, the actual results/data presented in this study is rather light and not very comprehensive. With more than 2000 samples, I would expect the authors to conduct a more thorough investigation of the phones and shoes microbiomes. For example, there is no mentioning of alpha-diversity or the taxonomic composition of the communities. Also, additional enrichment analyses (such as ANCOM and/or DeSeq2) may be used to complement Figure 3.

1.4 I humbly would also recommend a greater focus in the introduction section on the importance of undertaking the work described (e.g. the identification of the microbial dark matter and what that means for understanding phone and shoe microbiomes) from an ecological standpoint.

1.5 Please carefully review the referencing format of PeerJ. It appears that the journal names for some references do not conform to the PeerJ referencing format.

1.6 Can authors provide ethical approval statement?

Experimental design

2.1 The experimental design, including steps for read sequence quality control and filtering, is generally well thought-out and clear.

2.2 Various sections of the text now in the Results section are perhaps more suitable in the Materials and Methods section.

2.3 The authors indicate that 2,486 samples were included for sequencing. However, the SRA link provided on the manuscript only contains 2,234 files. Please clarify.

Validity of the findings

3.1 I feel that there is a general lack of statistical evidence to back the statements claimed by the authors. For example, the reason for not performing the ANOSIM analysis between the different sample group (when combining other studies in the meta-analysis) was not explained.

3.2 Results tend to agree their previous works but does not have immense novelty. The MDM results presented is promising but authors will need to elaborate both the analysis and discussion/significance of this tremendously.

Additional comments

Coil and colleagues describe a country-wide microbiome characterization of mobile phone and shoe surfaces, which appears to be a follow-up study of their previous works. The described investigation is generally interesting, but the study will benefit from a more comprehensive characterization of the phone/shoe microbiomes. There are instances where follow-up discussion is lacking, and because some text do not belong to the right sections, the manuscript feels disorganized and lacking focus. Also, references are lacking to support some of their claims. Overall, this manuscript definitely requires major revision, both in a sense that additional analyses and a re-structuring of the manuscript, before it can be accepted for publication. Specifically:

Line 99-100: please elaborate on “act in some ways.” In what ways?

Line 104-123: large chunks of this section should be in Materials and Methods. I am more interested here greater details about previous works conducted on indoor surfaces (especially phones and shoes). The authors do describe works conducted on phones and shoes, but should describe in more detail. There is also little information about the dark matter in which the authors have provided nice results later in the manuscript. Describing these points greater depth will also help with the comment directly below.

Line 123: from the introduction, the work seems to simply reflect on a repeat of the previous works conducted by the group, with a larger number of samples. Can the authors explicitly state what is currently lacking in the scientific field, and how the works presented help fill that knowledge gap (in addition to the statement that this work entails a larger sample size)?

Line 194: The first three paragraphs of the results section arguably should be in the Methods section. As a matter of fact, this first section of results/discussion would benefit from a general overview of the taxonomic composition and the microbial (alpha) diversity of the different surfaces.

Line 224: Please provide the ANOSIM R value to describe the strength of the significant clustering between the phone and shoe microbial communities.

Line 227: The authors have used the word "presumably." What is this presumption based on? I believe that this is based on previous publications. If that is the case, please cite relevant references.

Line 227-8: Please either perform the ANOSIM analysis here, or provide a valid reason for why this was not performed.

Line 237: fix sentence “Extracting VIMs not does not…”

Line 246-247: can the authors provide a description of the biogeography of BE microbiomes in general, and how their results here agree or disagree with previous publications? Also, the authors state that further works would reveal more information about biogeography, but why was this not performed and results included in the present manuscript? Also, please elaborate what kind of analysis is required in the future for the authors to understand “the influence of geographic location on microbial communities.” This phrase is very vague.

Line 247: missing punctuation at end of sentence.

Line 268/269/278: please provide references to support the conjectures.

Line 282-287: Not sure how this paragraph is related to the title of this section. This paragraph is perhaps better suited for either the intro or the methods section where sampling is described.

Line 294-300: the fact that the authors used “we focused” on these phyla suggests that there are in fact more phyla from the MDM detected, and the authors chose to analyze these ones further. If this is the case, explicitly explain why these were chosen. If this is the entire list of MDM phyla, rephrase this sentence to make less confusing.


Line 300-302: fix sentence "and the whether they"
Line 303-317: if the authors also performed enrichment analyses as recommended (ANCOM/DeSeq2), there would be a more statistical method in backing up claims suggesting that certain phyla are skewed towards phone/shoe samples.

Line 308: I presume these represent the prevalence of these MDMs in the samples. How abundant are these MDMs in each sample? Are these MDMs considered rare? If that is the case, are they rare throughout, or do they represent more abundant members is a particular subset of samples? Please provide more in-depth analyses of these MDMs.

Line 314: can the authors describe these prevalent phyla in detail, and how their ecological niche reflect on the results presented on phones/shoes?

Line 319: please provide reference to back up statement.

Line 318-320: what is the significance of this concluding sentence? Please elaborate and discuss.

Line 324-331: the conclusion right now simply regurgitates the findings presented in results section. What are the general significance and key of these results? What can we as readers learn from the results and its significance?

Line 340: please indicate metadata also available in the link (either here or in Materials and Methods section).

Table 1: please elaborate on “rough approximation.” Either provide more information here or on Materials and Methods section.

Table 1: will the authors be able to provide a supplementary table providing metadata on individual samples?

Reviewer 2 ·

Basic reporting

1. The following statement starting on line 113 “The second was to engage people in thinking about cell phones as being a proxy for sampling the microbes found on a person and their shoes as being a proxy for sampling the microbes found in a person’s environment” although true is a bit misleading in the context of this study. The authors did not collect skin or oral samples from the participants or environmental samples from the study sites, which prevented the authors from making objective connections from cell phones to owners and shoes to environment. The previous work by Lax et al. 2015, did collect samples from the owners, cell phones, floors, and environment, which permitted Lax et al., to determine if the cell phone was a proxy for the owner and the shoe a proxy for the environment.
2. The first three paragraphs of the Results/ Discussion section contain a mix of methods and results and discussion. These should be revised so that the appropriate content is found in the respective section.
3. In general, the results and discussion section is lacking in references to previous research to support many statements made. Lines 223, 244, 268, 273, and 278 are just a few examples. The authors may want to consider taking some time to review the literature to support arguments being made and to bolster the content of the results and discussion.
4. The following statement “the majority of people on the planet now possess a cell phone” found on line 96 needs a reference.

Experimental design

1. It was stated several times throughout the manuscript that ~3500 samples were initially collected. Although this is a laudable accomplishment, which should be noted, I fear that it could be misleading of the actual sample size of analyzed samples (N = ~2000 - 2500). The authors report that sequencing was performed on 2486 samples. However, in Table 1 the sum of samples collected at each site is only 2086. Based on the information in the manuscript there is no explanation of why there is a 400 sample difference between the table and the text.
2. The authors state that “some samples were lost in transit and a further 864 samples were excluded from sequencing due to an irretrievable loss of the sample ID data.” If this information is to be shared, it should be stated how many were lost in transit and why such a large number of samples lost their sample ID data. If the authors collected 3500 samples and excluded 864 from missing sample ID data (3500 – 864 = 2636) and had a sample size of 2486. Does that mean that 150 samples were “lost in transit” (2636 – 2486 = 150)? If that is the case, and the authors again choose to share this information, it should be explained how so many samples were lost in transit.

Validity of the findings

1. The authors reference visual associations and clustering within PCoA plots, and sometimes explicitly state that statistics were not run to support these visual trends (line 227). The authors have shown proficiency at running ANOSIM. All visual associations and trends should be supported by statistics.
2. The way that the centroids were determined for cell phones and shoes and how the line separating the centroids was made seems unconventional. Is this a validated method? Are there previous studies that have used this same method? If so, these studies should be referenced. If not, it would be more conventional to create a biplot.
3. The statement starting on line 225 is contradictory to arguments previously made in the manuscript. “Visual examination suggests that floor samples (light blue) group with shoes (as expected), while spacecraft (yellow) group with phones, presumably because both of these communities have major contribution from human associated taxa.” The authors previously argued that shoe samples carry taxa representative of the “environment” not human associated taxa.
4. Line 241 includes the following statement “The sport played at the venue where the sample was collected is less predictive…” It is not clear how this analysis was done because only 9 of the 38 venues listed in Table 1 can explicitly be associated to a single sport. It should be clearly outlined in the methods section how this analysis was done, and which venues were included.

Additional comments

1. The authors should consider revising the following statement starting on line 223 “this is supported by ANOSIM statistical analysis which showed a significance of 0.001 for this separation of shoes and phones.” to “this is supported by statistical analysis (ANOSIM; p = 0.001).”
2. There is a typo on line 237 “Extracting VIMs not does not include the optimization…”
3. There is a typo on line 288 “In relation to this, we examined how many (if any) of these microbes present in such…” "...present in such..." should be "...present on such..."
4. In Table 1 the authors may consider using the term “Juvenile” or something similar instead of “Kid.”
5. Figure 1 figure legend should define what “control”, “index”, and “other” sample types are.
6. In Figure 3 the dots should be colored by sample type rather than phyla.

·

Basic reporting

This paper proposes using cell phones and shoes as sampling devices for potential forensic use and epidemiology to probe the environmental and human microbiota. The authors use a very large-scale study to characterize these communities and conclude that shoe associated microbial community carries phyla typically found in environmental samples while the cell phones contain taxa abundant in the human microbiota. They also dive into microbial dark matter analysis and showed that it’s mainly detected in the microbial community associated with shoes.
1. The article is well written
2. Introduction and context are clear and relevant. A few minor comments about the introduction:
- line 96: “In addition to the millions of microbes that we carry around each day, the majority of people on the planet now possess a cell phone” this sentence seems incomplete.
- line 98: The main message of the paper cited from Meadow, Altrichter & Green, 2014 being that cell phone microbiota is representative of your index skin microbiome (“82% of the OTUs were shared between a person’s index and phone when considering the dominant taxa”). It would be worth emphasizing this previous finding in the introduction.
- line 120: In the method, you indicate that you also collected the date of sampling
-line 122: The introduction indicates that you collected ~3,500 samples, but you could only use ~2,500: it would be great to indicate this with the statement “To our knowledge, this represents the largest collection of bacterial community sequencing data associated with cell phones or shoes.”
In the discussion section, some references are missing:
- line 268 -> 278: several claims are made about taxa found and the community they’re found in. However, without any references.
- There are a lot of references cited for using Random Forest to extract features of importance: it’s unclear if the method used in the paper line 179->181. Did you use a specific package? Can you reference it?
- line 91: “This understanding has implications for fields ranging from medicine to forensics to architecture”. Please elaborate or cite a reference.
3. Structure is clear, but the results and discussion sections are not separated which is required in the instructions for the author section. This does not really pose any problem with the overall understanding of the paper of this paper, but I let the editor decide if it’s appropriate to keep it that way for peerJ.
4. Major comments:
- Figure 1 and 2: Were the data normalized?
- Figure 4 is also referenced as suppl 1.
- Table one is very long and needs to be either summarized or moved to supplemental information.
- The Labelling of the y-axis and the % represented in Figure 4 is unclear. Is it Predictive accuracy?
5. Yes, all data are available.

Experimental design

1. Yes
2. Yes
3. Yes
4. Some major comments here:
- The DADA2 workflow is well described and reproducible (ideally the authors would make the code available, but they did make the intermediate files available). However, there is not enough information available about the random forest approach used: Line 179->181. The authors need to indicate the package they used for their example and the relevant literature specific to this package; not a general citation about Random Forest.
- line 197: Provide the mean of reads/datasets with standard deviation rather than the total number of reads.
- line 201: Can you indicate the number of samples per datasets (439 total)? And as indicated above, the mean of the number of read/dataset with standard deviation.
- line 217: Can you provide the number of ASVs at each of the filtering steps (and again the mean for each dataset rather than the total of reads)?
- line 222: Did you normalized your data? Same question line 23, did you normalize the ASV count?

Minor comments here (which could be avoided if you were to provide the used code):
- Line 151: What Q2 score (value) did you use? If the code is not available, you need to indicate which threshold you used. I’m assuming this is the two expected error (indicated below?) maxEE in DADA2 “filterAndTrim” function.
- Line 153: This statement is confusing with the Q2 score indicate above.
- Line 156-157: “Error models were calculated using one million reads for the three published data sets” as a minimum number of total bases to use for error rate learning in DADA2?
- Line 161: 7 table sequences or 4 (from the four datasets)?
- Line 162: Using the function tip_glom: id you use h=0.2 (default)?
- Line 237 remove the "not"

Validity of the findings

The impact of this study is well assessed, and the sample size definitively makes the rational believable. However, I would like to draw the attention of the author on several analyses that need to be improved for this paper to be published.
- line 165: You decided to remove the un-identified phyla from your analysis. But, the analysis displayed figure 1 is based on microbial structure distances, regardless of the phylogenetic assignment of the ASV. Can you draw the same conclusion if you consider all the ASVs, i.e. are the two communities are distinct?
- line 217: Do you observe any difference between the sequencing depth of the four different datasets? Or in the number of ASVs detected? While the sampling method is similar, the matrix from which it is sampled (shoes with soil particles vs cell phones) it drastically different and could result in very different DNA yield. In the same vein, it would also be great to comment on beta diversity between the matrix.
- line 228: You visually comment to which centroid the dataset considered is the closest to. Could you test this (by looking into distance to each considered centroid)?
- line 244: You suggest “local biogeography effects in patterning the microbial community”. The claim is unclear to me: beyond the random forest analysis, is the community actually predictive of the sampling location? This would need to be tested and discuss.
- line 256-262: Do you have any statistics supporting these associations? While I agree that some of the ASVs seem to be on one side or the other of the line separating the two centroids, there are quite a few tools to confirm that taxa are more abundant in one sample type than the other (Deseq, MetaGenomSeq) (Love et al., 2014) (Paulson et al., 2013), or by comparing distance to the centroid in each plot.
REFERENCES:
Love MI, Huber W, Anders S. Moderated estimation of fold change and dispersion for RNA-seq data with DESeq2. Genome Biol. 2014;15(12):550–21. PMCID: PMC4302049
Paulson JN, Stine OC, Bravo HC, Pop M. Differential abundance analysis for microbial marker-gene surveys. Nat. Methods. BioMed Central; 2013 Sep 29;10(12):1200–2.

Additional comments

The conclusions are well stated, but the authors need to address a few critical points as indicated above (especially regarding batch effects and confounding factors) to support the claims of this paper.

---

## Round 0.2 · Minor Revisions

This manuscript was substantially improved with this revised version. However, I agree with the reviewer that it is critical for the work to be reproducible. Please pay special attention to the missing link for the code repository, and the random forest methods.

Reviewer 1 ·

Basic reporting

The revised version now has sufficient and appropriate referencing and the background on the need to look at the microbial dark matter is described in the introduction.

Experimental design

The methods section has now been elaborated as recommended.

Validity of the findings

Statistical tests and results have now been provided throughout the manuscript.

Additional comments

The authors have addressed the previous comments, and have substantially improved the manuscript. I approve this manuscript for publication.

·

Basic reporting

The authors addressed my concerns in this section.
Minor comment : Line 207 “The figure was generated “: Consider adding the final figure number.

Experimental design

The authors addressed my concerns in this section.

Minor comment:
Initial Comment: The DADA2 workflow is well described and reproducible (ideally the authors would make the code available, but they did make the intermediate files available). However, there is not enough information available about the random forest approach used: Line 179->181. The authors need to indicate the package they used for their example and the relevant literature specific to this package; not a general citation about Random Forest.

First Rebuttal answer: “Language has been added to the text to make this more clear. The code is now all present in the linked repository, it may have been missing in the original submission.”

New comment: The level of details in the method section is now suitable for publication, but I still can’t find the code repository link in the method section (maybe you’ve submitted it somewhere else online and it’s not in the PDF itself?)

Validity of the findings

The authors addressed my concerns in this section.

Additional comments

Great job revising the manuscript!

---

## Round 0.3 · accepted · Accept

Thank you for your attention to detail in preparing this manuscript. I like the idea of objects as sources and sinks of human microbiome diversity, so glad to see it published.